# SCHRÖDINGER BRIDGE TO BRIDGE GENERATIVE DIFFUSION METHOD TO OFF-POLICY EVALUATION

## ABSTRACT

The problem of off-policy evaluation (OPE) in reinforcement learning (RL), which evaluates a given policy using data collected from a different behavior policy, plays an important role in many real-world applications. The OPE under the model of episodic non-stationary finite-horizon Markov decision process (MDP) has been widely studied. However, the general model-free importance sampling (IS) methods suffer from the curse of horizon and dimensionality, while the improved marginal importance sampling (MIS) can only be restrained to the case where the state space $\mathcal{S}$ is sufficiently small. The model-based methods often have limited scope of application. To find a widely-applicable OPE algorithm when $\mathcal{S}$ is continuous and high-dimensional that avoids the curse of horizon and dimensionality, which means the error of the estimator grows exponentially with the number of horizon $H$ and the dimension $d$ of the state space $\mathcal{S}$, we apply the diffusion Schr"odinger bridge generative model to construct a model-based estimator (CDSB estimator). Moreover, we established the statistical rate of the estimation error of the value function with a polynomial rate of $O(H^2\sqrt{d})$, which, to the best of our knowledge, is one of the first theoretical rate results on applying Schr"odinger bridge to reinforcement learning. This breaks the restraint of the complexity of the state space for OPE under MDP with large horizon and can be applied to various real-life decision problems with continuous setting, which is shown in our simulation using our method in continuous, high-dimensional and long-horizon RL environments and its comparison with other existing algorithms.

## 1 INTRODUCTION

The problem of off-policy evaluation (OPE) in reinforcement learning is evaluating the average return value of a given unknown policy (referred to as the target policy) leveraging data gathered from a distinct behavior policy. Given the increasing need for OPE in domains like self-driving and healthcare, the development of efficient algorithms for off-policy evaluation has emerged as a critical priority.

Of all the OPE problems, OPE under the setting of Markov decision process (MDP) is of great importance. For MDP-setting OPE problems, there are various, both model-free and model-based algorithms in the literature. For model-free algorithms, the method of importance sampling (IS) is the most representative and serves as an efficient bridge between the target policy and behavior policy and is widely used for short-horizon OPE problems. (Precup et al., 2000; Hanna et al., 2018; Robins et al., 2000) However, the traditional IS algorithm as well as many other model-free algorithm (for example, Kallus & Uehara (2020)) suffers from the curse of horizon, which means the MSE of IS estimator grows exponentially with the number of horizon $H$. (Liu et al., 2020; Jiang & Li, 2016; Precup et al., 2000; Thomas et al., 2015; Farajtabar et al., 2018; Guo et al., 2017; Thomas & Brunskill, 2016) Xie et al. (2019) proposes the Marginal Importance Sampling (MIS) estimator, reducing the dependence of the number of horizons to polynomial. However, the applicability of the MIS estimator is limited to the case where the state space $\mathcal{S}$ is sufficiently small and discrete. Uehara et al. (2020) employs minimax optimization to avoid curse of horizon and dimensionality, however it is generally challenging to compute. It necessitates additional properties, such as the Q-function of the MDP belonging to a Reproducing Kernel Hilbert Space (RKHS) function class, to ensure the effectiveness of minimax optimization.

There are also many model-based methods for MDP-setting OPE problems where the transition functions of the MDP system are directly estimated. (Liu et al., 2018; Gottesman et al., 2019; Hallak et al., 2015) Some model-based estimators can efficiently avoid the curse of horizon and work well in the case that the state space is continuous. However, a common problem with model-based estimators is that they usually require sharp conditions on the transition and policy functions, which in turn results in a relatively small coverage of the MDP setting of the OPE problem. For example, The model-based approach discussed in Uehara & Sun (2021), which focuses on continuous state spaces, mandates policy functions to belong to a finite function class due to the PAC-learning bound incorporating the term.

Generally speaking, there hasn't been a practical algorithm for MDP-setting off-policy evaluation that can be applied to scenarios where state space $\mathcal{S}$ is sufficiently large and avoids the curse of horizon and dimensionality at the same time, while covers a wide range of MDP settings

In deep learning, a generative model describes how a dataset is generated, which empowers the generation of a substantial volume of data that conforms to a desired distribution possible, even if the target distribution is in a very complex space. This intrinsic capability renders generative modeling highly relevant and applicable in the context of distribution estimation. (Liu et al., 2021; Chen et al., 2019; Liang, 2021; Li et al., 2019; Abbasnejad et al., 2019; Zhang et al., 2020; Liang, 2018). In recent studies, the methodology of diffusion and score-matching is widely used in generative modeling to solve problems in image synthesis and data recovery. (Ho et al., 2020; Hyvärinen, 2005; Song & Ermon, 2020; Song et al.; Vahdat et al., 2021; Jo et al., 2022; Dockhorn et al.; Wang et al., 2023; Janner et al., 2022), Moreover, recent studies (Wang et al., 2021; De Bortoli et al., 2021; Winkler et al., 2023; Shi et al., 2023) view the classical Schrödinger bridge problem (Rüschendorf & Thomsen, 1993) revised under the methodologies of machine learning (Vargas, 2021; Pavon et al., 2021) as a generative modeling problem and uses score-based diffusion to find solutions for Schrödinger bridge problem.

To tackle the problem that conventional density estimators cannot handle complex state and action space, in this paper we implement the methodology of diffusion Schrödinger bridge to directly estimate the transition functions and construct a model-based estimator (the CDSB estimator). The idea of using generative model as transition function estimator in RL, to our knowledge, has not been discovered in the literature. In comparison of Xie et al. (2019), our approach avoids the curse of horizon, meanwhile it is applicable for OPE problems in continuous and high-dimensional space. In comparison of Uehara et al. (2020) and Uehara & Sun (2021), our approach covers a wider range of MDP settings, as it does not impose the requirement for MDP functions to belong to specific function classes; it solely necessitates boundedness and smoothness of transition and policy functions.

Previous studies have discussed the convergence rate and asymptotic properties of the solution to Schrödinger bridge, most of which based on the iterative propotional fitting (IPF) method of solving the Schrödinger bridge. (Deligiannidis et al., 2021; Gibbs & Su, 2002) Instead, our paper apply the likelihood training method to solve the diffusion Schrödinger bridge as in Chen et al. (2023b) and Chen et al. (2023c). To derive the convergence rate under this method, we take advantage of the score-matching error estimation in Chen et al. (2023a) and derive an total-variation error bound using Girsanov's theorem, which is the first likelihood training Schrödinger bridge error bound in the literature. With this error bound, we ultimately derive an $O(H^2\sqrt{d})$-bound of absolute-value error for the estimation of the value function $V^\pi$ under an assumption of universal score estimation error.

**Contributions.** We conclude our main contributions as follows. First, we introduce the diffusion Schrödinger bridge generative model for density estimation and design an applicable algorithm to adapt such estimator in model-based off-policy evaluation, therefore extending solveability of OPE problems to the setting of high-dimensional and complex state and action space. Second, we prove the quantitative statistical convergence rate for diffusion Schrödinger bridge solved by likelihood training in total variance norm. Third, we bound the absolute value (1-norm) error of our model-based value function estimator , which has a $O(H^2\sqrt{d})$ convergence rate. To the best of our knowledge, this is the first quantitative convergence result employing diffusion Schrödinger bridge into the context of reinforcement learning.

## 1.1 RELATED WORK

**Off-Policy-Evaluation** In reinforcement learning, Off Policy Evaluation refers to accurately evaluating a target policy using previously logged feedback data of a behavior policy (Dudík et al.,

2014). Importance sampling (IS) and marginal importance sampling (MIS) estimators are widely used for OPE problems. (Precup et al., 2000; Hanna et al., 2018; Robins et al., 2000; Xie et al., 2019) Kostrikov & Nachum (2020) uses self-normalized step-wise importance sampling for the problem. Le et al. (2019) trains a neural network to estimate the value of the evaluation policy $\pi$ by bootstrapping from $Q(s', \pi(s'))$. Model-based methods are also adopted as in the work of Zhang et al., Liu et al. (2018), Gottesman et al. (2019) and Hallak et al. (2015). Uehara et al. (2020) uses minimax optimization to solve the problem which performs well in continuous state space. A more thorough review of the literature on OPE can be found in Uehara et al. (2022).

**Schrödinger Bridge Problem** The SB problem is an entropy-regularized Optimal Transport problem introduced by Schrödinger (1932). Genevay et al. (2018) deals with SB problem in the context of discrete distribution. Finlay et al. (2020) solves SB problem by approximating the SB solution by a diffusion whose drift is computed using potentials. Another prevalent method for solving SB is using Iterative Proportional Fitting which is also adopted in De Bortoli et al. (2021) to formulate a generative model for faster generation. The convergence results for IPF have been resolved under classical compactness assumptions as in Chen et al. (2016).

## 2 PROBLEM FORMULATION

**Symbols and notations.** We consider the problem of offline policy evaluation for a finite horizon MDP, which is defined by $M = (\mathcal{S}, \mathcal{A}, T, R, H)$, where $\mathcal{S}$ is a continuous state space, $\mathcal{A}$ a continuous action space, $T_t : \mathcal{S} \times \mathcal{A} \times \mathcal{S} \to [0, 1]$ is the transition function with $T_t(s'|s, a)$ defined by probability of transitioning into state $s'$ upon taking action $a$ in state $s$ at time $t$, and $R_t : \mathcal{S} \times \mathcal{A} \to \mathbb{R}$ is the reward function. $R_t(s, a)$ is the deterministic immediate reward associated with taking action $a$ in state $s$ at time $t$, and $H$ denotes the finite horizon. Without loss of generality, we study the case where $\mathcal{S} = \mathcal{A} = [0, 1]^d \subset \mathbb{R}^d, d \geq 1$. We use $\Pr\{E\}$ and $\mathbb{E}\{E\}$ to denote the probability and expectation of an event $E$, $\mathbb{E}\{E|F\}$ to denote the conditional expectation of event $E$ given the condition $F$. Denote $[n]$ to be the set of natural numbers $\{1, \cdots, n\}$. Use $\mathcal{P}(p_1, p_2)$ to denote the set of all path measures on $\mathcal{S}$ throughout time interval $[0, T]$ with $p_1$ and $p_2$ as its marginal densities at $t = 0$ and $T, n \in \mathbb{N}$. Denote the Kullback-Leibler divergence between $p$ and $q$ to be $\mathrm{KL}(p|q)$, and denote the total-variation norm between $p$ and $q$ to be $\mathrm{TV}(p, q)$. For a random variable $\mathbf{X}$ with probability density $\mathbf{p}$, for a map $f$, we denote $f_\#\mathbf{q}$ the probability density of random variable $f(\mathbf{X})$.

Let $\mu, \pi$ be policies whose output is a distribution of actions given an observed state. Make $\mu$ the behavioral policy and $\pi$ the target policy. Denote $\mu(a|s)$ the probability density function of actions given state. Moreover, we denote $d_t^\pi(s_t)$ the induced state distribution by $\pi$ at time $t$. When $t = 1$, the initial distributions are known and identical $d_1^\pi = d_0$. For $t > 1$, $d_t^\pi(s_t)$ is defined recursively as follows:

$$d_t^\pi(s_t) = \int_{\mathcal{S}} P_t^\pi(s_t|s_{t-1}) d_{t-1}^\pi(s_{t-1}),$$

$$\text{where } P_t^\pi(s_t|s_{t-1}) = \int_{\mathcal{A}} T_t(s_t|s_{t-1}, a_{t-1}) \pi(a_{t-1}|s_{t-1}) \mathrm{d}a_{t-1}.$$

**Problem setup.** The key to offline policy evaluation is to find an estimator $\widehat{V}^\pi$ using the data collected by the behavior policy $\mu$ and the known action probabilities to estimate the value function

$$V^\pi = \sum_{t=1}^{H} \int_{\mathcal{A}} \int_{\mathcal{S}} d_t^\pi(s_t) \pi(a_t|s_t) R_t(s_t, a_t) \mathrm{d}s_t \mathrm{d}a_t,$$

where we assume $\pi(a|s)$ and $\mu(a|s)$ is known for all $(s, a) \in \mathcal{S} \times \mathcal{A}$, $R_t(s_t, a_t)$ is unknown. The transition distributions $T_t(s_t|s_{t-1}, a_{t-1})$ is unknown and not easy to be observed.

Different from various previous studies in this field such as (Xie et al., 2019), which focus on the case where $\mathcal{S}$ and $\mathcal{A}$ is discrete and low-dimensional, we provide an estimator $\widehat{V}^\pi$ under the condition that $\mathcal{S}$ and $\mathcal{A}$ is high-dimensional and continuous. In particular, we set $\mathcal{S} = \mathcal{A} = [0, 1]^d, d \geq 1$. Our main strategy is constructing model-based estimators, that is, directly estimating the transition function $T_t(s_t|s_{t-1}, a_{t-1})$.

# 3 MODEL-BASED CONDITIONAL DIFFUSION SCHRÖDINGER BRIDGE ESTIMATOR

To construct model-based estimators for OPE problem, one has to provide reliable estimation $\widehat{T}_t(s_t|s_{t-1}, a_{t-1})$ of the transition function $T_t(s_t|s_{t-1}, a_{t-1})$ for all $t = 1, \cdots, H$. Consequently, we get an estimator for the value function for any given target policy $\pi$:

$$\widehat{V}^\pi = \sum_{t=1}^{H} \int_{\mathcal{A}} \int_{\mathcal{S}^t} \widehat{R}_t(s_t, a_t)\pi(a_t|s_t)\widehat{P}_t^\pi(s_t|s_{t-1})\cdots\widehat{P}_2^\pi(s_2|s_1)d_0(s_1)\mathrm{d}s_1\cdots\mathrm{d}s_t\mathrm{d}a_t, \quad (1)$$

where

$$\widehat{P}_t^\pi(s_t|s_{t-1}) = \int_{\mathcal{A}} \widehat{T}_t(s_t|s_{t-1}, a_{t-1})\pi(a_{t-1}|s_{t-1})\mathrm{d}a_{t-1}, \ t = 2, \cdots, H, \quad (2)$$

and $\widehat{R}_t(s_t, a_t)$ being estimation of the reward function.

In our work, we will construct the estimation $\widehat{T}_t(s_t|s_{t-1}, a_{t-1})$ using conditional diffusion Schrödinger bridge to get our estimator $\widehat{V}^\pi$ as above.

## 3.1 SCHRÖDINGER BRIDGE PROBLEM FOR DENSITY ESTIMATION

The classical Schrödinger Bridge problem (Föllmer, 1988) in continuous time setting aims to find a path measure on time interval $[0, T]$ that achieves a minimum Kullback-Leibler divergence relative to a reference density under given marginal conditions, that is, to find $Q^\star \in \mathcal{P}(p_{\mathrm{data}}, p_{\mathrm{prior}})$ such that

$$Q^\star = \mathrm{argmin}\{\mathrm{KL}(Q|P) : Q \in \mathcal{P}(p_{\mathrm{data}}, p_{\mathrm{prior}})\}, \quad (3)$$

where $P \in \mathcal{P}_{N+1}$ is a reference path measure on $\mathcal{S}$ in $[0, T]$ that can be designed, $p_{\mathrm{data}}$ is the target distribution we aim to estimate, $p_{\mathrm{prior}}$ is a known prior distribution. Suppose that $Q^\star$ is available, then the target distribution $p_{\mathrm{data}}$ can be generated by $Q^\star$ using the known prior distribution $p_{\mathrm{prior}}$ and $Q^\star$, which means we can achieve density estimation of $p_{\mathrm{data}}$ by solving the Schrödinger bridge problem 3.

If we set the reference density $P$ as the path measure of the add-noise SDE in score-based generative modeling, which is

$$\mathrm{d}\mathbf{X}_r = f(\mathbf{X}_r, r)\mathrm{d}r + g(r)\mathrm{d}\mathbf{W}_r, \ \mathbf{X}_0 \sim p_{\mathrm{data}}, r \in [0, T], \quad (4)$$

where $f(\cdot, r) : \mathbb{R}^n \to \mathbb{R}^n, g(t) \in \mathbb{R}$ are the drift and diffusion, and $\mathbf{W}_r \in \mathbb{R}^n$ is the standard Brownian process. Then we get the diffusion Schrödinger bridge. We denote $f(\mathbf{X}_r, r) \equiv f$ and $g(r) \equiv g$ for simplicity.

For the diffusion Schrodinger bridge, the optimality condition 3 can be characterized by two PDEs that are coupled through boundary conditions. The result is summarized as below.

**Theorem 3.1.1**(Chen et al., 2021; Pavon & Wakolbinger, 1991; Caluya & Halder, 2021) Let $\Psi(r, \boldsymbol{x})$ and $\widehat{\Psi}(r, \boldsymbol{x})$ be the solutions to the following PDEs:

$$\begin{cases} \frac{\partial \Psi}{\partial x} = -\nabla_{\boldsymbol{x}}\Psi^\top f - \frac{1}{2}\mathrm{Tr}(g^2\nabla_{\boldsymbol{x}}^2\Psi) \\ \frac{\partial \widehat{\Psi}}{\partial x} = -\nabla_{\boldsymbol{x}} \cdot (\widehat{\Psi}f) + \frac{1}{2}\mathrm{Tr}(g^2\nabla_{\boldsymbol{x}}^2\widehat{\Psi}) \end{cases} s.t. \Psi(0, \cdot)\widehat{Psi}(0, \cdot) = p_{\mathrm{data}}, \Psi(T, \cdot)\widehat{\psi}(T, \cdot) = p_{\mathrm{prior}}. \quad (5)$$

Then, the solution to the optimization 3 can be expressed by the path measure of the forward SDE

$$\mathrm{d}\mathbf{X}_r = [f + g^2\nabla_{\boldsymbol{x}} \log \Psi(r, \mathbf{X}_r)]\mathrm{d}r + g\mathrm{d}\mathbf{W}_r, \ \mathbf{X}_0 \sim p_{\mathrm{data}} \quad (6)$$

or equivalently the backward SDE

$$\mathrm{d}\mathbf{X}_r = [f - g^2\nabla_{\boldsymbol{x}} \log \widehat{\Psi}(r, \mathbf{X}_r)]\mathrm{d}r + g\mathrm{d}\mathbf{W}_r, \ \mathbf{X}_T \sim p_{\mathrm{prior}}, \quad (7)$$

So finding the solution to the diffusion Schrödinger bridge problem is equivalent to finding solutions $\Psi(r, \boldsymbol{x})$ and $\widehat{\Psi}(r, \boldsymbol{x})$ to PDE 5.

### 3.2 SOLVING SCHRÖDINGER BRIDGE USING LIKELIHOOD TRAINING

Denote $\mathbf{Z}_r = g\nabla_{\boldsymbol{x}} \log \Psi$ and $\widehat{\mathbf{Z}}_r = g\nabla_{\boldsymbol{x}} \log \widehat{\Psi}$. Then the set $(\mathbf{Z}_r, \widehat{\mathbf{Z}}_r)$ contains all the information of the diffusion Schrödinger bridge (DSB) model by the above analysis. Suppose $q_r$ is the marginal distribution at time $r \in [0, T]$ of the solution to the diffusion Schrödinger bridge problem 3, then the log-likelihood of a data point $\boldsymbol{x}_0$ from $p_{\text{data}}$ generated by the diffusion Schrödinger bridge is, by definition, $\log q_0(\boldsymbol{x}_0)$. We have the following theorem.

**Theorem 3.2.1**(Chen et al., 2023b) The log-likelihood of the DSB model $(\mathbf{Z}_r, \widehat{\mathbf{Z}}_r)$ at data point $\boldsymbol{x}_0$ can be expressed as

$$\log q_0(\boldsymbol{x}_0) = \mathbb{E}[\log q_T(\mathbf{X}_T)] - \int_0^T \mathbb{E}[\frac{1}{2}\|\mathbf{Z}_r\|^2 + \frac{1}{2}\left\|\widehat{\mathbf{Z}}_r\right\|^2 + \nabla_{\boldsymbol{x}} \cdot (g\widehat{\mathbf{Z}}_r - f) + \widehat{\mathbf{Z}}_r^\top \mathbf{Z}_r]\mathrm{d}t.$$

Consequently, we can maximize $\mathcal{L}_{SB}(\boldsymbol{x}_0; \theta, \phi)$, which shares the same expression as $\log q_0(\boldsymbol{x}_0)$ above with $\mathbf{Z}_r \approx \mathbf{Z}(r, \boldsymbol{x}; \theta)$ and $\widehat{\mathbf{Z}}_r \approx \widehat{\mathbf{Z}}(r, \boldsymbol{x}; \theta)$ are approximated by parameterized models, in order to solve the DSB problem. By Theorem 11 of Chen et al. (2023b), using the symmetric property of the Schrödinger bridge, we can convert maximizing $\mathcal{L}_{SB}(\boldsymbol{x}_0; \theta, \phi)$ to maximizing the following two objectives:

$$\tilde{\mathcal{L}}_{SB}(\boldsymbol{x}_0; \phi) = -\int_0^T \mathbb{E}_{\mathbf{X}_r \sim 6}[\frac{1}{2}\left\|\widehat{\mathbf{Z}}(r, \mathbf{X}_r; \phi)\right\|^2 + g\nabla_{\boldsymbol{x}}\widehat{\mathbf{Z}}(r, \mathbf{X}_r; \phi) + \mathbf{Z}_r^\top\widehat{\mathbf{Z}}(r, \mathbf{X}_r; \phi)]\mathrm{d}r, \quad (8)$$

$$\tilde{\mathcal{L}}_{SB}(\boldsymbol{x}_T; \theta) = -\int_0^T \mathbb{E}_{\mathbf{X}_r \sim 7}[\frac{1}{2}\|\mathbf{Z}(r, \mathbf{X}_r; \theta)\|^2 + g\nabla_{\boldsymbol{x}}\mathbf{Z}(r, \mathbf{X}_r; \theta) + \widehat{\mathbf{Z}}_r^\top\mathbf{Z}(r, \mathbf{X}_r; \theta)]\mathrm{d}r. \quad (9)$$

### 3.3 CONDITIONAL LIKELIHOOD TRAINING

The most straightforward way to apply DSB to our model-based OPE estimator is to construct a diffusion Schrödinger bridge with target distribution $p_{\text{data}}(s_t) = T_t(s_t|s_{t-1}, a_{t-1})$ for each $t \in \{2, \cdots, H\}$ and each $(s_{t-1}, a_{t-1}) \in \mathcal{S} \times \mathcal{A}$, which is not computational achievable when $\mathcal{S}$ and $\mathcal{A}$ are continuous. Instead, we view $T_t(s_t|s, a)$ as a conditional probability density function conditioned on parameter $(t, s, a)$, which can further be included in the training parameters as $\tilde{\phi} = (\phi, t, s, a)$ and $\tilde{\theta} = (\theta, t, s, a)$. Chen et al. (2023c) provide a practical algorithm implementation using a conditional mask (see Section 5.2 of Chen et al. (2023c)) , which is an alternate training of the following loss with masks,

$$\tilde{\mathcal{L}}_{SB}(\boldsymbol{x}_0; \phi) = -\int_0^T \mathbb{E}_{\mathbf{X}_r \sim 6}[\frac{1}{2}\left\|\widehat{\mathbf{Z}}(r, \mathbf{X}_r; \phi) \circ \mathbf{M}\right\|^2 + g\nabla_{\boldsymbol{x}}[\widehat{\mathbf{Z}}(r, \mathbf{X}_r; \phi) \circ \mathbf{M}]$$
$$+ [\mathbf{Z}_r \circ \mathbf{M}]^\top[\widehat{\mathbf{Z}}(r, \mathbf{X}_r; \phi) \circ \mathbf{M}]\mathrm{d}r, \quad (10)$$

,

where $\mathbf{M}$ is the target mask that has element 1 for the target index and 0 otherwise.

Meanwhile, in order to empirically generate data from SDEs, in practice we will make discretization for the time interval $[0, T]$. An $N$-step discretization is to divide $[0, T]$ into $[kh, (k+1)h], k = 0, \cdots, N-1$, where the step size $h := \frac{T}{N}$.

Using the conditional maximum likelihood training of the DSB problem, we finally get the estimation $\widehat{T}_t(s_t|s_{t-1}, a_{t-1})$ of the transition function $T_t(s_t|s_{t-1}, a_{t-1})$ for all $t = 2, \cdots, H$ and $(s_t, s_{t-1}, a_{t-1}) \in \mathcal{S} \times \mathcal{S} \times \mathcal{A}$, which we use to construct our OPE estimator by Equation 1 and Equation 2. We call our estimator the Conditional Diffusion Schrödinger Bridge (CDSB) estimator.

In implementation, $\mathbf{X_0}$ is $(s_{t-1}, a_{t-1}, s_t)$. We stack them to be a longer vector. And the conditional masks will take element 1 on the index of $s_t$. Besides, we will also train a neural network for reward function $\widehat{R}_t(s_t, a_t)$ which takes state and action as input to predict the reward. The detailed algorithm for training and OPE evaluation are summarised in algorithm 1

---

**Algorithm 1:** CDSB Estimator Training and OPE

---

**Training:**

**Input:** Sampler $p_{prior}$ and $p_{obs}$, fixed condition-target masks $\mathbf{M}$

**Output:** Trained backward policy $\widehat{Z}(r, \tilde{\phi})$

**for** *k in 1:K* **do**

    **Repeat**:

    Sample $\mathbf{X}_{r\in[0,T]}$ following 6 where $x_0 \sim p_{obs}$.

    Compute $\tilde{\mathcal{L}}_{SB}(\boldsymbol{x}_0; \phi)10$ using masks $\mathbf{M}$.

    Take gradient and update parameter $\phi$.

    Sample $\mathbf{X}_{r\in[0,T]}$ following 7 where $X_T \sim p_{prior}$.

    Compute $\tilde{\mathcal{L}}_{SB}(\boldsymbol{x}_T; \theta)$.

    Take gradient and update parameter $\theta$.

**end**

\# Use output $\widehat{Z}(r, \tilde{\phi})$ and masks $\mathbf{M}$ to form a conditional sampler $\widehat{T}(s_t|s_{t-1}, a_{t-1}, t)$ where $(s_{t-1}, a_{t-1})$ is condition and $s_t$ is target. Conditional generation is done following equation 7.

**Model-based OPE:**

**Input:** Target policy $\pi$, sampled initial states $\{s_0^{(i)}\}_{i=1}^n$, trained conditional sampler $\widehat{T}(s_t|s_{t-1}, a_{t-1}, t)$, trained reward network $\widehat{R}$

**Output:** $\widehat{V}^\pi$

**for** *t in 1:H* **do**

    \# Sample $\{a_t^{(i)}\}_{i=1}^n$ from $\pi$

    Sample $\{s_t^{(i)}\}_{i=1}^n$ from $\widehat{T}$.

    Predict $\{r_t^{(i)}\}_{i=1}^n$ using reward network $\widehat{R}$.

**end**

\# Compute $\widehat{V}^\pi = \frac{1}{n} \sum_{i=1}^n \sum_{t=1}^H r_t^{(i)}$

---

# 4 THEORETICAL ANALYSIS OF THE CDSB ESTIMATOR

In this section, we provide the approximation property of the CDSB estimator. To get a convergent result, the Schrödinger bridge model derived from the MDP model, the parameterized model estimation error and target policies $\pi$ require the following assumptions:

1. $\Psi(r, \boldsymbol{x})$ and $\widehat{\Psi}(r, \boldsymbol{x})$ in Section 3.1 satisfies that $\nabla_{\boldsymbol{x}} \log \Psi(r, \boldsymbol{x})$ and $\nabla_{\boldsymbol{x}} \log \widehat{\Psi}(r, \boldsymbol{x})$ are $L$-Lipschitz with respect to variable $\boldsymbol{x}$ for all $r \in [0, T]$.

2. For all $t \in \{2, \cdots, H\}$ and all $(s, a) \in \mathcal{S} \times \mathcal{A}$, $\mathbb{E}_{\mathbf{X} \sim T_t(\cdot|s,a)} \|\mathbf{X}\|^2 \leq m^2 < \infty$.

3. The drift $f$ and the diffusion $g$ in Equation 4 satisfies: $f$ has a finite upper bound $M < +\infty$, $g(r) \equiv c$ is a constant function with $0 < c \leq M$.

4. The unknown reward function $R_t(s_t, a_t)$ has a uniform upper bound $R_{\max} = \sup_{s_t, a_t, t} R_t(s_t, a_t)$ with respect to all $t = 1, \cdots, H$.

5. For target policy $\pi$, $\tau := \sup_{s \in \mathcal{S}, a \in \mathcal{A}} |\pi(a|s)| < \infty$.

6. for all $k = 1, \cdots, N$, all $t = 1, \cdots, H$, all $(s, a) \in \mathcal{S} \times \mathcal{A}$,

$$\mathbb{E}_{q_{kh,t,s,a}}[\|\mathbf{Z}(kh, \mathbf{X}_{kh}, (\theta, t, s, a)) - \mathbf{Z}_{kh}\|^2] \leq \epsilon^2,$$

$$\mathbb{E}_{q_{kh,t,s,a}}[\|\widehat{\mathbf{Z}}(kh, \mathbf{X}_{kh}, (\phi, t, s, a)) - \widehat{\mathbf{Z}}_{kh}\|^2] \leq \epsilon^2, \quad |\widehat{R}_t(s, a) - R_t(s, a)|^2 \leq \epsilon^2,$$

    where $q_{kh,t,s,a}$ is the marginal density at time $kh \in [0, T]$ of the solution to the DSB 3 with $p_{\text{data}} = T_t(\cdot|s, a)$.

Assumption (4) is easily achievable, since an upper bound for reward function is guaranteed in almost every reinforcement learning problem. Assumption (5) (boundedness of the target policy $\pi$) also covers most off-policy evaluation problems. Assumption (2) requires a second moment bound of the transition function. Since in our setting, $\mathcal{S} = [0, 1]^d$ is bounded and $\text{supp}\{T_t(\cdot|s, a)\} \in \mathcal{S}$ for all $t = 2, \cdots, H$ and $(s, a) \in \mathcal{S} \times \mathcal{A}$, this assumption naturally holds in our setting. Assumption (3) is also easily achievable since both drift and diffusion can be designed. In practice, we can apply the standard denoising diffusion probabilistic modeling (DDPM) setting $f(t, \mathbf{X}_t) = -\mathbf{X}_t$ (bounded since $\mathbf{X}_t$ is bounded) and $g(t) = \sqrt{2}$. Assumption (1) requires lipschitzness of $\nabla_{\boldsymbol{x}} \log \Psi(r, \boldsymbol{x})$ and $\nabla_{\boldsymbol{x}} \log \widehat{\Psi}(r, \boldsymbol{x})$, which could be derived from the lipschitzness and lower-boundedness of $p_{\text{data}} =$

$T_t(\cdot|s,a)$ by analysis of the parabolic PDE 5. Meanwhile, the lipschitzness and lower-boundedness of the transition function is a conventional setting in continuous MDP system. The final assumption (6) is an score estimation error assumption, which is similar to the assumption in Lee et al. (2022). Notice that our assumption requires the learning error $\epsilon$ uniformly on all $t = 2, \cdots, H$ and $(s,a) \in \mathcal{S} \times \mathcal{A}$, which is still an realistic assumption under the algorithm of conditional likelihood training.

**Theorem 4.1** Under Assumptions (1)-(6), let $\widehat{V}^\pi$ be the output of CDSB estimator, and suppose that the step size $h := \frac{T}{N}$ satisfies $h \lesssim \frac{1}{L}$, where $L \geq 1$. Suppose the diffusion time $T \geq \max\{1, \frac{1}{\tau^2}\}$, then it holds that

$$|\widehat{V}^\pi - V^\pi| \lesssim R_{\max}\tau^2 H^2(\epsilon + M^3 L^{3/2}T\sqrt{dh} + LMmh)\sqrt{T}. \tag{11}$$

We make a few remarks about the above theorem. Firstly, the error bound $|\widehat{V}^\pi - V^\pi|$ only has a 2-order polynomial dependence on the number of horizon $H$, which shows that the CDSB estimator avoids the exponential curse of horizon in comparison with traditional IS estimators (Liu et al., 2020). On the other hand, the bound of error requires only a $\sqrt{d}$-dependence on the dimension $d$ of the state space $\mathcal{S}$, which indicates that our algorithm also avoids the curse of dimensionality, which means it has excellent performance on continuous and high-dimensional state and action space. Finally, The error bound can be easily controlled by narrowing the estimation error $\epsilon$ and the diffusion step size $h$, which are both easy to achieve during practical empirical computation.

To prove the above theorem, we need to compare the structure of $V^\pi$ and $\widehat{V}^\pi$. Noticing that

$$V^\pi = \sum_{t=1}^{H} \int_{\mathcal{A}} \int_{\mathcal{S}^t} R_t(s_t, a_t)\pi(a_t|s_t)P_t^\pi(s_t|s_{t-1})\cdots P_2^\pi(s_2|s_1)d_0(s_1)\mathrm{d}s_1\cdots\mathrm{d}s_t\mathrm{d}a_t,$$

and

$$\widehat{V}^\pi = \sum_{t=1}^{H} \int_{\mathcal{A}} \int_{\mathcal{S}^t} \widehat{R}_t(s_t, a_t)\pi(a_t|s_t)\widehat{P}_t^\pi(s_t|s_{t-1})\cdots \widehat{P}_2^\pi(s_2|s_1)d_0(s_1)\mathrm{d}s_1\cdots\mathrm{d}s_t\mathrm{d}a_t.$$

It comes naturally that a uniform bound of $\int_{S} |\widehat{P}_t^\pi(s_t|s_{t-1}) - P_t^\pi(s_t|s_{t-1})|\mathrm{d}s_t$ on all $t = 2, ..., H$ and all $s_{t-1} \in \mathcal{S}$ can be used to bound $|\widehat{V}^\pi - V^\pi|$.

Since $\widehat{P}_t^\pi(s_t|s_{t-1}) = \int_{\mathcal{A}} \widehat{T}_t(s_t|s_{t-1}, a_{t-1})\pi(a_{t-1}|s_{t-1})\mathrm{d}a_{t-1}$ and $P_t^\pi(s_t|s_{t-1}) = \int_{\mathcal{A}} T_t(s_t|s_{t-1}, a_{t-1})\pi(a_{t-1}|s_{t-1})\mathrm{d}a_{t-1}$ and $\pi$ is upper-bounded with $\tau$, we only require a uniform bound of $\int_{S} |\widehat{T}_t(s_t|s_{t-1}, a_{t-1}) - T_t(s_t|s_{t-1}, a_{t-1})|\mathrm{d}s_t$ on all $t = 2, \cdots, H$ and all $(s_{t-1}, a_{t-1}) \in \mathcal{S} \times \mathcal{A}$, which is guaranteed in the following theorem:

**Theorem 4.2** For any $t = 2, \cdots, H$ and any $(s_{t-1}, a_{t-1}) \in \mathcal{S} \times \mathcal{A}$, suppose the diffusion time $T \geq \max\{1, \frac{1}{\tau^2}\}$, we have

$$\mathrm{TV}(\widehat{T}_t(\cdot|s,a), T_t(\cdot|s,a)) \lesssim (\epsilon + M^3 L^{3/2}T\sqrt{dh} + LMmh)\sqrt{T}.$$

This theorem is proved mainly using the Girsanov's theorem. The method is similar to Chen et al. (2023a), with some alternations under the diffusion Schrödinger bridge setting. With Theorem 4.2 proved, we are able to prove Theorem 4.1 using some iterations on $t$.

## 5 EXPERIMENTS

### 5.1 SETTING AND RESULT

We conduct our experiments on the DeepMind control suite (Tassa et al., 2018), a set of control tasks implemented in MuJoCo (Todorov et al.). We use a subset of the offline datasets from RL Unplugged (Gulcehre et al., 2020), the details of which are provided in table 1. These environments capture a wide range of complexity, from 40K transitions in a 5-dimensional cartpole environment to 1.5 million transitions on complex manipulation tasks. We follow part of the evaluation protocol in the Deep OPE benchmark(Fu et al., 2020).

As for the policies, we adopt the policy trained by Kostrikov & Nachum (2020) for each task as behavior policies. Offline datasets are generated following such policies. Four different level of noise

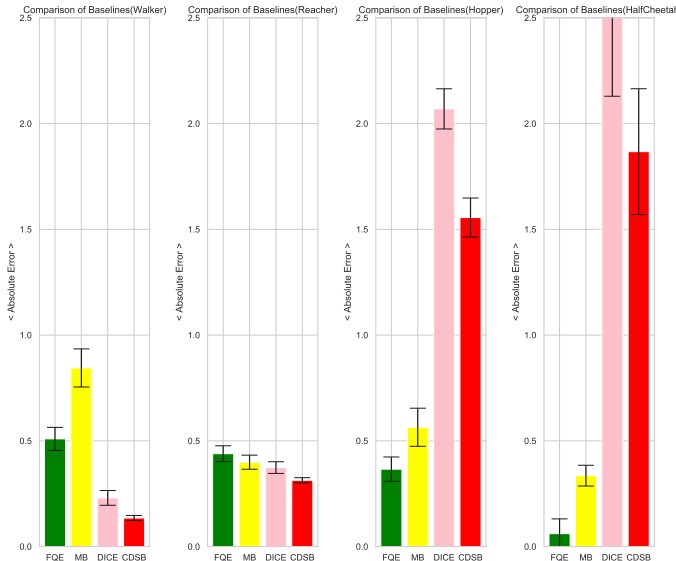

Figure 1: Mean Abosolute Error with Error Bar

Table 1: Summary of the offline datasets used

|  | Reacher | Hopper | HalfCheetah | Walker |
|---|---|---|---|---|
| State dim. | 11 | 11 | 17 | 17 |
| Action dim. | 2 | 3 | 6 | 6 |
| Number of episodes | 1M | 1M | 1M | 1M |
| Infinite horizon | yes | yes | yes | yes |

is added to the behavior policies to form target policies. The evaluation is done by performing OPE on different behavior-target policy pairs for each task. After that, absolute error is measured for each OPE problem, and median absolute error is used to evaluate the performance of an OPE algorithm on a task. We compare our method(CDBS) with the following baseline: **Fitted Q-Evaluation(FQE)**, **Model-Based**, **DICE**. These baselines include model-based and model-free method. We follow the implementation of these baselines in Kostrikov & Nachum (2020).

The summary statistic is displayed in table 2. Our method **achieves state-of-the-art performance on two among four OPE tasks** measured by median absolute error. We also provide the result of the mean absolute error with error bar in figure 1 to show robustness of each method.

## 5.2 CONDITIONAL GENERATION DETAILS

In this section, we briefly describe the pipeline of the conditional diffusion schrodinger bridge network. More details about the neural networks, training procedure, inference, baseline models, and evaluation can be found in Appendix.

As described in section 3.3, we use two separate neural networks to model the forward or backward policy. The backward network needs to handle partially observed input and conduct conditional inference. More specifically, the backward policy has format $\widehat{Z}(r, \mathbf{X}_r, \mathbf{M}, \phi)$ which takes in diffusion time, condition masks, and outputs the policy of the whole time window (its outputs at condition positions are usually ignored). While the forward network, as an assistant for training the backward policy, does not need to process partial input, and we use a modified U-Net as the neural network(Ronneberger et al., 2015). In both networks, the diffusion time is incorporated through embedding. Similar to the design Tashiro et al. (2021), the backward policy handles the input with irregular conditions based on the transformer, where the condition information is encoded through channel concatenation, feature index embedding, and time index embeddings .

Table 2: OPE Evaluation Result

| Median Absolute Error | Reacher | Hopper | HalfCheetah | Walker |
|---|---|---|---|---|
| FQE | 0.374 | 0.096 | **0.218** | 0.232 |
| MB | 0.336 | **0.064** | 0.286 | 0.781 |
| Dual Dice | 0.417 | 2.595 | 1.032 | 0.201 |
| **CDSB(ours)** | **0.318** | 1.0405 | 1.276 | **0.080** |

## 6 CONCLUSIONS

In this paper, we propose the CDSB estimator to solve off-policy evaluation under finite-horizon MDP with continuous and high-dimensional state space $\mathcal{S}$. In comparison with traditional model-based approaches and classic model-free approaches such as importance sampling, our approach avoids the curse of horizon and dimensionality with only polinomial dependence on horizon $H$ and dimension $d$, making it possible to solve OPE problem efficiently under the complex state space $\mathcal{S}$. Meanwhile, our estimator proves efficient under a wide range of MDP settings since it solely requires boundedness and smoothness of transition and policy functions.

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

# A APPENDIX

## A.1 PROOF OF THEOREM 4.2

The key to proving Theorem 4.2 is the use of Girsanov's theorem.

**Lemma 1**(Girsanov's theorem) For $t \in [0, T]$, let $\mathcal{L}_t = \int_0^t b_s \mathrm{d}B_s$ where $B$ is a $Q$-Brownian motion. Assume $\mathbb{E}_Q \int_0^T \|b_s\|^2 \, \mathrm{d}s < \infty$. Then, $\mathcal{L}$ is a $Q$-martingale in $L^2(Q)$. Moreover, if

$$\mathbb{E}_Q \mathcal{E}(\mathcal{L})_T = 1, where \mathcal{E}(\mathcal{L})_t := \exp\left(\int_0^t b_s \mathrm{d}B_s - \frac{1}{2}\int_0^t \|b_s\|^2 \, \mathrm{d}s\right),$$

then $\mathcal{E}(\mathcal{L})$ is also a $Q$-martingale and the process

$$t \mapsto B_t - \int_0^t b_s \mathrm{d}s$$

is a Brownian motion under $P := \mathcal{E}_T Q$, the probability distribution with density $\mathcal{E}(\mathcal{L})_T$ w.r.t. Q.

In the proof below, for any fixed $t \in \{2, \cdots, H\}$ and $(s, a) \in \mathcal{S} \times \mathcal{A}$, let $p_{\mathrm{data}} = T_t(\cdot|s, a)$, we denote the path measure of the backward SDE 7 and forward SDE 6 (they share the same solution) to be $Q_T := Q_T(\cdot|t, s, a)$. Denote the path measure generated from the conditional likelihood training to be $P_T := P_T(\cdot|t, s, a, \phi, \theta)$. Denote $\widehat{\mathbf{Z}}(\cdot, \cdot, \tilde{\theta}) := \widehat{\mathbf{Z}}$ and $\mathbf{Z} := \mathbf{Z}(\cdot, \cdot, \tilde{\phi})$. By Assumption $(1)\sim(6)$, the following analysis holds for any given $t = 2, \cdots, H$ and $(s_{t-1}, a_{t-1}) \in \mathcal{S} \times \mathcal{A}$.

**Theorem 4.2** For any $t = 2, \cdots, H$ and any $(s_{t-1}, a_{t-1}) \in \mathcal{S} \times \mathcal{A}$, suppose the diffusion time $T \geq \max\{1, \frac{1}{\tau^2}\}$, we have

$$\mathrm{TV}(\widehat{T}_t(\cdot|s, a), T_t(\cdot|s, a)) \lesssim (\epsilon + M^3 L^{3/2} T\sqrt{dh} + LMmh)\sqrt{T}.$$

*Proof.* We start by proving

$$\sum_{k=0}^{N-1} \mathbb{E}_{Q_T} \int_{kh}^{(k+1)h} \left\|\widehat{\mathbf{Z}}(kh, \mathbf{X}_{kh}) - c\nabla_{\boldsymbol{x}} \log \widehat{\Psi}(r, \mathbf{X}_r)\right\|^2 \mathrm{d}r \lesssim (\epsilon^2 + M^6 L^3 dh + M^2 h^2 m^2)T.$$

For $r \in [kh, (k+1)h]$, we can decompose

$$\mathbb{E}_{Q_T}[\left\|\widehat{\mathbf{Z}}(kh, \mathbf{X}_{kh}) - c\nabla_{\boldsymbol{x}} \log \widehat{\Psi}(r, \mathbf{X}_r)\right\|^2]$$

$$\lesssim \mathbb{E}_{Q_T}[\left\|\widehat{\mathbf{Z}}(kh, \mathbf{X}_{kh}) - c\nabla_{\boldsymbol{x}} \log \widehat{\Psi}(kh, \mathbf{X}_{kh})\right\|^2]$$

$$+ \mathbb{E}_{Q_T}[\left\|g\nabla_{\boldsymbol{x}} \log \widehat{\Psi}(kh, \mathbf{X}_{kh}) - g\nabla_{\boldsymbol{x}} \log \widehat{\Psi}(r, \mathbf{X}_{kh})\right\|^2]$$

$$+ \mathbb{E}_{Q_T}[\left\|g\nabla_{\boldsymbol{x}} \log \widehat{Psi}(r, \mathbf{X}_{kh}) - g\nabla_{\boldsymbol{x}} \log \widehat{Psi}(r, \mathbf{X}_r)\right\|^2]$$

$$\lesssim \epsilon^2 + \mathbb{E}_{Q_T} \left\|g\nabla_{\boldsymbol{x}} \log\left(\frac{\widehat{\Psi}(kh, \mathbf{X}_{kh})}{\widehat{\Psi}(r, \mathbf{X}_{kh})}\right)\right\|^2 + M^2 L^2 \mathbb{E}_{Q_T} \|\mathbf{X}_{kh} - \mathbf{X}_r\|^2$$

Notice that if $S : \mathbb{R}^d \to \mathbb{R}^d$ is the mapping $S(x) = \exp(-(r - kh))x$, then $\widehat{\Psi}(T - kh, \cdot) = S(\widehat{\Psi}(T-r, \cdot) * \mathcal{N}(0, 1-\exp(-2(r-kh))))$. We can use Lemma 2 with $\alpha = \exp(r-kh) = 1+O(h)$ and $\sigma^2 = 1 - \exp(-2(r - kh)) = O(h)$ and obtain

$$\mathbb{E}_{Q_T} \left\|g\nabla_{\boldsymbol{x}} \log\left(\frac{\widehat{\Psi}(kh, \mathbf{X}_{kh})}{\widehat{\Psi}(r, \mathbf{X}_{kh})}\right)\right\|^2 \lesssim M^2(L^2 dh + L^2 h^2 \|\mathbf{X}_{kh}\|^2 + L^2 h^2 \left\|\nabla_{\boldsymbol{x}} \log \widehat{\Psi}(r, \mathbf{X}_{kh})\right\|^2).$$

Also we have

$$\left\|\nabla_{\boldsymbol{x}}\widehat{Psi}(r,\mathbf{X}_{kh})\right\|^2 \leq \left\|\nabla_{\boldsymbol{x}}\log\widehat{\Psi}(r,\mathbf{X}_r)\right\|^2 + \left\|\nabla_{\boldsymbol{x}}\log\widehat{Psi}(r,\mathbf{X}_{kh}) - \nabla_{\boldsymbol{x}}\log\widehat{\Psi}(r,\mathbf{X}_r)\right\|^2$$

$$\leq \left\|\nabla_{\boldsymbol{x}}\log\widehat{\Psi}(r,\mathbf{X}_r)\right\|^2 + L^2\left\|\mathbf{X}_{kh}-\mathbf{X}_r\right\|^2.$$

So

$$\mathbb{E}_{Q_T}[\left\|\widehat{\mathbf{Z}}(kh,\mathbf{X}_{kh}) - c\nabla_{\boldsymbol{x}}\log\widehat{\Psi}(r,\mathbf{X}_r)\right\|^2]$$

$$\lesssim \epsilon^2 + M^2(L^2dh + L^2h^2\mathbb{E}_{Q_T}\left\|\mathbf{X}_{kh}\right\|^2 + L^2h^2\mathbb{E}_{Q_T}\left\|\nabla_{\boldsymbol{x}}\log\widehat{\Psi}(T-r,\mathbf{X}_r)\right\|^2 + L^2\mathbb{E}_{Q_T}\left\|\mathbf{X}_{kh}-\mathbf{X}_r\right\|^2).$$

Using $L$-smoothness of $\nabla_{\boldsymbol{x}}\log\widehat{\Psi}$ and $\nabla_{\boldsymbol{x}}\log\Psi$, by (Vempala & Wibisono (2019), Lemma 9) and (Chen et al. (2023a), Lemma 10) , we have

$$\mathbb{E}\left\|\nabla_{\boldsymbol{x}}\log\widehat{\Psi}(r,X_r)\right\|^2 \leq Ld,$$

and

$$\mathbb{E}\left\|\nabla_{\boldsymbol{x}}\log\Psi(r,X_r)\right\|^2 \leq Ld.$$

On the other hand, for $0 \leq s < r$, by the forward process 6, we have

$$\mathbb{E}_{Q_T}\left\|\mathbf{X}_r - \mathbf{X}_s\right\|^2 = \mathbb{E}_{Q_T}[\left\|\int_s^r (f + c^2\nabla_{\boldsymbol{x}}\log\Psi(r,\mathbf{X}_r))\mathrm{d}r + c(B_r - B_s)\right\|^2]$$

$$\lesssim (r-s)\int_s^r \mathbb{E}\left\|f + c^2\nabla_{\boldsymbol{x}}\log\Psi(r,\mathbf{X}_r)\right\|^2 \mathrm{d}r + M(r-s)d$$

$$\lesssim (r-s)^2M^2 + (r-s)^2M^4Ld + M(r-s)d$$

As a result, we get

$$\mathbb{E}\left\|\mathbf{X}_{kh}\right\|^2 \leq \mathbb{E}\left\|\mathbf{X}_0\right\|^2 + T^2M^2 + T^2M^4Ld + MTd$$

$$\leq m^2 + T^2M^2 + T^2M^4Ld + MTd$$

and

$$\mathbb{E}\left\|\mathbf{X}_{kh} - \mathbf{X}_r\right\|^2 \leq h^2M^2 + h^2M^4Ld + Mhd.$$

Combining the results above, we get that

$$\mathbb{E}_{Q_T}[\left\|\widehat{\mathbf{Z}}(kh,\mathbf{X}_{kh}) - c\nabla_{\boldsymbol{x}}\log\widehat{\Psi}(r,\mathbf{X}_r)\right\|^2]$$

$$\lesssim \epsilon^2 + M^2[L^2dh + L^2h^2(m^2 + T^2M^2 + T^2M^4Ld + MTd) + L^2h^2Ld + L^2(h^2M^2 + h^2M^4Ld + Mhd)]$$

$$\lesssim \epsilon^2 + M^6L^3T^2dh + M^2L^2h^2m^2.$$

(Suppose $T \geq 1$ and $h \lesssim \frac{1}{L}$) So we have

$$\sum_{k=0}^{N-1}\mathbb{E}_{Q_T}\int_{kh}^{(k+1)h}\left\|\widehat{\mathbf{Z}}(kh,\mathbf{X}_{kh}) - c\nabla_{\boldsymbol{x}}\log\widehat{\Psi}(r,\mathbf{X}_r)\right\|^2 \mathrm{d}r \lesssim (\epsilon^2 + M^6L^3T^2dh + M^2L^2h^2m^2)T.$$

Now we apply an approximation argument to use Girsanov's theorem and prove Theorem 4.2.

For $r \in [0,T]$, let $\mathcal{L}_r = \int_0^r b_s\mathrm{d}B_s$ where $B$ is a $Q_T$-Brownian motion. For $r \in [kh,(k+1)h]$, define

$$b_r = -c\nabla_{\boldsymbol{x}}\log\widehat{\Psi}(r,\mathbf{X}_r) + \widehat{\mathbf{Z}}(kh,\mathbf{X}_{kh}).$$

From above,

$$\mathbb{E}_{Q_T}\int_0^T \left\|b_s\right\|^2 \mathrm{d}s \lesssim (\epsilon^2 + M^6L^3T^2dh + M^2L^2h^2m^2)T < \infty,$$

using (Le Gall (2016), Proposition 5.11), $(\mathcal{E}(\mathcal{L})_r)_{r\in[0,T]}$ (see the definition in Lemma 1) is a local martingale (see Definition 1). Therefore, there exists a non-decreasing sequence of stopping

time $T_n \uparrow T$ such that $(\mathcal{E}(\mathcal{L})_{r \wedge T_n})_{r \in [0,T]}$ is a martingale. Notice that $\mathcal{E}(\mathcal{L})_{r \wedge T_n} = \mathcal{E}(\mathcal{L}_r^n)$ where $\mathcal{L}_r^n = \mathcal{L}_{r \wedge T_n}$. Since $\mathcal{E}(\mathcal{L}_r^n)_{r \in [0,T]}$ is a martingale, we have

$$\mathbb{E}_{Q_T} \mathcal{E}(\mathcal{L}^n)_T = \mathbb{E}_{Q_T} \mathcal{E}(\mathcal{L}^n)_0 = 1,$$

so that $\mathbb{E}_{Q_T} \mathcal{E}(\mathcal{L})_{T_n} = 1$.

Apply Girsanov's theorem to $\mathcal{L}_r^n = \int_0^r b_s \mathbf{1}_{[0,T_n]}(s) \mathrm{d}B_s$ where $B$ is a $Q_T$-Brownian motion and get that under $P^n := \mathcal{E}(\mathcal{L})_T Q_T$, there exists a Brownian motion $\beta^n$ such that for $r \in [0,T]$,

$$dB_r = \left[ -c \nabla_{\boldsymbol{x}} \log \widehat{\Psi}(r, \mathbf{X}_r) + \widehat{\mathbf{Z}}(kh, \mathbf{X}_{kh}) \right] \mathbf{1}_{[0,T_n]}(r) \mathrm{d}r + \mathrm{d}\beta_r^n.$$

By the backward SDE 7, under $Q_T$ we have

$$\mathrm{d}\mathbf{X}_r = -[f - c^2 \nabla_{\boldsymbol{x}} \log \widehat{\Psi}(r, \mathbf{X}_r)] \mathrm{d}r + c \mathrm{d}B_r, \ \mathbf{X}_0 \sim p_{\mathrm{prior}}.$$

The equation still holds $P^n$-a.s. since $P^n \ll Q_T$. Combining the two equations above then we obtain that $P^n$-a.s.,

$$\mathrm{d}\mathbf{X}_r = \left[ -f + c\widehat{\mathbf{Z}}(kh, \mathbf{X}_{kh}) \right] \mathbf{1}_{[0,T_n]}(r) \mathrm{d}r + \left[ -f + c^2 \nabla_{\boldsymbol{x}} \log \widehat{\Psi}(T-r, \mathbf{X}_r) \right] \mathbf{1}_{[T_n,T]}(r) \mathrm{d}r + c \mathrm{d}\beta_r^n, \ \mathbf{X}_0 \sim p_{\mathrm{prior}}.$$

i.e. path measure $P^n$ is the solution to the above SDE. So we have

$$\mathrm{KL}(Q_T | P^n) = \mathbb{E}_{Q_T} \log \mathcal{E}(\mathcal{L})_{T_n}^{-1} = \mathbb{E}_{Q_T}[-\mathcal{L}_{T_n} + \frac{1}{2} \int_0^{T_n} \|b_s\|^2 \, \mathrm{d}s] = \mathbb{E}_{Q_T} \frac{1}{2} \int_0^{T_n} \|b_s\|^2 \, \mathrm{d}s$$

$$\leq \mathbb{E}_{Q_T} \frac{1}{2} \int_0^T \|b_s\|^2 \, \mathrm{d}s \lesssim (\epsilon^2 + M^6 L^3 T^2 dh + M^2 L^2 h^2 m^2)T$$

where we used that $\mathbb{E}_{Q_T} \mathcal{L}_{T_n} = 0$ because $\mathcal{L}$ is a $Q_T$-martingale and $T_n$ is a bounded stopping time.(Le Gall (2016), Corollary 3.23)

Consider a coupling of $(P^n)_{n \in \mathbb{N}}, P_T$: a sequence of stochastic process $(\mathbf{X}^n)_{n \in \mathbb{N}}$ over the same proability space, a stochastic process $\mathbf{X}$ and a single Brownian motion $W$ over the same space s.t.

$$\mathrm{d}\mathbf{X}_r^n = \left[ -f + c\widehat{\mathbf{Z}}(kh, \mathbf{X}_{kh}^n) \right] \mathbf{1}_{[0,T_n]}(r) \mathrm{d}r + \left[ -f + c^2 \nabla_{\boldsymbol{x}} \log \widehat{\Psi}(T-r, \mathbf{X}_r^n) \right] \mathbf{1}_{[T_n,T]}(r) \mathrm{d}r + c \mathrm{d}W_r,$$

$$\mathrm{d}\mathbf{X}_r = \left[ -f + c\widehat{\mathbf{Z}}(kh, \mathbf{X}_{kh}) \right] \mathrm{d}r + c \mathrm{d}W_r,$$

$$\mathbf{X}_0 = \mathbf{X}_0^n \sim p_{\mathrm{prior}}.$$

By definition of $P^n$ and $P_T$, the distribution of $\mathbf{X}^n$ ($\mathbf{X}$) is $P^n$ ($P_T$).

Let $\delta > 0$ and consider the map $\pi_\delta : \mathcal{C}([0,T]; \mathbb{R}^d) \to \mathcal{C}([0,T]; \mathbb{R}^d)$ defined by

$$\pi_\delta(\omega)(r) := \omega(r \wedge (T - \delta)).$$

Notice that $\mathbf{X}_r^n = \mathbf{X}_r$ for every $r \in [0,T_n]$, using Lemma 3, we have $\pi_\delta(\mathbf{X}^n) \to \pi_\delta(\mathbf{X})$ a.s., uniformly over $[0,T]$. Therefore, $\pi_{\delta\#} P^n \to \pi_{\delta\#} P_T$ weakly. Using the lower semicontinuity of the KL divergence and the data-processing inequality (Amb (2005), Lemma 9.4.3 and Lemma 9.4.5), we get

$$\mathrm{KL}((\pi_\delta)_\# Q_T | (\pi_\delta)_\# P_T) \leq \liminf_{n \to \infty} \mathrm{KL}((\pi_\delta)_\# Q_T | (\pi_\delta)_\# P^n)$$

$$\leq \liminf_{n \to \infty} \mathrm{KL}(Q_T | P^n)$$

$$\lesssim (\epsilon^2 + M^6 L^3 T^2 dh + M^2 L^2 h^2 m^2)T.$$

Finally, using Lemma 4, $\pi_\delta(\omega) \to \omega$ as $\delta \to 0$ uniformly over $[0,T]$. Therefore, using (Amb (2005), Corollary 9.4.6), $\mathrm{KL}((\pi_\delta)_\# Q_T | (\pi_\delta)_\# P_T) \to \mathrm{KL}(Q_T | P_T)$ as $\delta \to 0$. Since the marginal distribution at $T = 0$ of $Q_T$ is $T_t(\cdot | s, a)$ and the marginal distribution at $T = 0$ of $P_T$ is $\widehat{T}_t(\cdot | s, a)$, by data processing inequality we ultimately have

$$\mathrm{KL}(T_t(\cdot | s, a) | \widehat{T}_t(\cdot | s, a)) \lesssim (\epsilon^2 + M^6 L^3 T^2 dh + M^2 L^2 h^2 m^2)T.$$

We conclude the proof using Pinsker's inequality ($\mathrm{TV}^2 \leq \mathrm{KL}$). $\qquad \square$

### A.2 Proof of Theorem 4.1

In this section, we give the proof of Theorem 4.1, which is our main theorem.

**Theorem 4.1** Under Assumptions (1)-(6), let $\widehat{V}^\pi$ be the output of CDSB estimator, and suppose that the step size $h := \frac{T}{N}$ satisfies $h \lesssim \frac{1}{L}$, where $L \geq 1$. Suppose the diffusion time $T \geq \max\{1, \frac{1}{\tau^2}\}$, then it holds that

$$|\widehat{V}^\pi - V^\pi| \lesssim R_{\max}\tau^2 H^2(\epsilon + M^3 L^{3/2}T\sqrt{dh} + LMmh)\sqrt{T}. \tag{12}$$

*Proof.* We have

$$V^\pi = \sum_{t=1}^H \int_{\mathcal{A}} \int_{\mathcal{S}^t} R_t(s_t, a_t)\pi(a_t|s_t)P_t^\pi(s_t|s_{t-1})\cdots \widehat{P}_2^\pi(s_2|s_1)d_0(s_1)\mathrm{d}s_1\cdots\mathrm{d}s_t\mathrm{d}a_t,$$

and

$$\widehat{V}^\pi = \sum_{t=1}^H \int_{\mathcal{A}} \int_{\mathcal{S}^t} \widehat{R}_t(s_t, a_t)\pi(a_t|s_t)\widehat{P}_t^\pi(s_t|s_{t-1})\cdots \widehat{P}_2^\pi(s_2|s_1)d_0(s_1)\mathrm{d}s_1\cdots\mathrm{d}s_t\mathrm{d}a_t.$$

By Theorem 4.2, assumption (6) and the definition of total-variation norm, for all $s \in \mathcal{S}$ and all $t \in \{2, \cdots, T\}$, we have

$$\int_{\mathcal{S}} |P_t^\pi(s'|s) - \widehat{P}_t^\pi(s'|s)|\mathrm{d}s' = \int_{\mathcal{S}} |\int_{\mathcal{A}} \pi(a|s)(T_t(s'|s,a) - \widehat{T}_t(s'|s,a))\mathrm{d}a|\mathrm{d}s$$

$$\lesssim \tau(\epsilon + M^3 L^{3/2}T\sqrt{dh} + LMmh)\sqrt{T} =: \delta_0,$$

$$\int_{\mathcal{A}} |\widehat{R}_t(s,a) - R_t(s,a)|\mathrm{d}a \leq \epsilon \lesssim \delta_0,$$

since $T \geq \max\{1, \frac{1}{\tau^2}\}$.

So

$$|\widehat{V}^\pi - V^\pi|$$

$$\leq \tau \sum_{t=1}^H \left| \left| \int_{\mathcal{A}} \int_{\mathcal{S}^t} \widehat{R}_t(s_t, a_t)\widehat{P}_t^\pi(s_t|s_{t-1})\cdots \widehat{P}_2^\pi(s_2|s_1)d_0(s_1)\mathrm{d}s_1\cdots\mathrm{d}s_t\mathrm{d}a_t - \right. \right.$$

$$\left. \left. \int_{\mathcal{A}} \int_{\mathcal{S}^t} R_t(s_t, a_t)P_t^\pi(s_t|s_{t-1})\cdots P_2^\pi(s_2|s_1)d_0(s_1)\mathrm{d}s_1\cdots\mathrm{d}s_t\mathrm{d}a_t \right| \right.$$

$$\leq \tau \sum_{t=1}^H \int_{\mathcal{A}} \int_{\mathcal{S}^t} \left| \widehat{R}_t(s_t, a_t)\widehat{P}_t^\pi(s_t|s_{t-1})\cdots \widehat{P}_2^\pi(s_2|s_1)d_0(s_1) - R_t(s_t, a_t)P_t^\pi(s_t|s_{t-1})\cdots P_2^\pi(s_2|s_1)d_0(s_1) \right| \mathrm{d}s_1\cdots\mathrm{d}s_t\mathrm{d}a_t$$

$$\leq \tau \sum_{t=1}^H \int_{\mathcal{A}} \int_{\mathcal{S}^t} \left( \left| \left(\widehat{R}_t(s_t, a_t) - R_t(s_t, a_t)\right) \widehat{P}_t^\pi(s_t|s_{t-1})\cdots \widehat{P}_2^\pi(s_2|s_1)d_0(s_1) \right| \right.$$

$$\left. + \left| R_t(s_t, a_t) \left(\widehat{P}_t^\pi(s_t|s_{t-1})\cdots \widehat{P}_2^\pi(s_2|s_1) - P_t^\pi(s_t|s_{t-1})\cdots P_2^\pi(s_2|s_1)\right) d_0(s_1) \right| \right) \mathrm{d}s_1\cdots\mathrm{d}s_t\mathrm{d}a_t$$

$$\cdots$$

$$\leq \tau \sum_{t=1}^H \left( \int_{\mathcal{A}} \int_{\mathcal{S}^t} \left| \widehat{R}_t(s_t, a_t) - R_t(s_t, a_t) \right| \left| P_t^\pi(s_t|s_{t-1}) \right| \left| P_{t-1}^\pi(s_{t-1}|s_{t-2}) \right| \cdots \left| P_2^\pi(s_2|s_1)d_0(s_1) \right| \mathrm{d}s_1\cdots\mathrm{d}s_t\mathrm{d}a_t \right.$$

$$\left. + \cdots + \int_{\mathcal{A}} \int_{\mathcal{S}^t} \left| \widehat{R}_t(s_t, a_t) - R_t(s_t, a_t) \right| \left| \widehat{P}_t^\pi(s_t|s_{t-1}) - P_t^\pi(s_t|s_{t-1}) \right| \cdots \left| \widehat{P}_2^\pi(s_2|s_1) - P_2^\pi(s_2|s_1) \right| d_0(s_1)\mathrm{d}s_1\cdots\mathrm{d}s_t\mathrm{d}a_t \right)$$

The summation above contains $2^{t-1} - 1$ items, each term $|\cdot|$ in the integration of each item is either $|\widehat{P}_j^\pi(s_j|s_{j-1}) - P_j^\pi(s_j|s_{j-1})|$ $(|\widehat{R}_t(s_t, a_t) - R_t(s_t, a_t)|)$ or $|P_j^\pi(s_j|s_{j-1})|$ $(|R_t(s_t, a_t)|)$ for $j = 2, \cdots, t$, but not all $|P_t^\pi(s_j|s_{j-1})|$. Relax all the $|\widehat{P}_j^\pi(s_j|s_{j-1}) - P_j^\pi(s_j|s_{j-1})|$ and $|\widehat{R}_t(s_t, a_t) -$

$R_t(s_t, a_t)|$ to their uniform upper bound (with respect to $s_{j-1}$ and $s_t$) $\delta_0$. Since $P_j^\pi$ are non-negative for $t = 1, \cdots, t-1$, the terms of each item in the summation are then relaxed to

$$\delta_0^{t-1-k} \int_{\mathcal{A}} \int_{\mathcal{S} \times \cdots \times \mathcal{S}} R_t(s_t, a_t) P_{j_k}^\pi(s_{j_k}|s_{j_k-1}) \cdots P_{j_1}^\pi(s_{j_1}|s_{j_1-1}) d_0(s_1) \mathrm{d}s_t \cdots \mathrm{d}s_1 \mathrm{d}a_t,$$

or

$$\delta_0^{t-k} \int_{\mathcal{S} \times \cdots \times \mathcal{S}} R_t(s_t, a_t) P_{j_k}^\pi(s_{j_k}|s_{j_k-1}) \cdots P_{j_1}^\pi(s_{j_1}|s_{j_1-1}) d_0(s_1) \mathrm{d}s_t \cdots \mathrm{d}s_1,$$

where $1 \le k \le t-1$, $j_1 < \cdots < j_k$ and $\{j_1, \cdots, j_k\} \in \{2, \cdots, t\}$. By the definition of $P_j^\pi$, it's easy to verify that

$$\int_{\mathcal{S}^t} P_{j_k}^\pi(s_{j_k}|s_{j_k-1}) \cdots P_{j_1}^\pi(s_{j_1}|s_{j_1-1}) d_0(s_1) \mathrm{d}s_t \cdots \mathrm{d}s_1 = 1$$

and

$$\int_{\mathcal{A}} \int_{\mathcal{S}^t} R_t(s_t, a_t) P_{j_k}^\pi(s_{j_k}|s_{j_k-1}) \cdots P_{j_1}^\pi(s_{j_1}|s_{j_1-1}) d_0(s_1) \mathrm{d}s_t \cdots \mathrm{d}s_1 \mathrm{d}a_t \le R_{\max}$$

for any $1 \le k \le t-1$, $j_1 < \cdots < j_k$ and $\{j_1, \cdots, j_k\} \in \{2, \cdots, t\}$. So that the summation

$$\int_{\mathcal{A}} \int_{\mathcal{S}^t} \left| \widehat{R}_t(s_t, a_t) - R_t(s_t, a_t) \right| \left| \widehat{P}_t^\pi(s_t|s_{t-1}) \right| \left| \widehat{P}_{t-1}^\pi(s_{t-1}|s_{t-2}) \right| \cdots \left| \widehat{P}_2^\pi(s_2|s_1) d_0(s_1) \right| \mathrm{d}s_1 \cdots \mathrm{d}s_t \mathrm{d}a_t$$

$$+ \cdots + \int_{\mathcal{A}} \int_{\mathcal{S}^t} \left| \widehat{R}_t(s_t, a_t) - R_t(s_t, a_t) \right| \left| \widehat{P}_t^\pi(s_t|s_{t-1}) - P_t^\pi(s_t|s_{t-1}) \right| \cdots \left| \widehat{P}_2^\pi(s_2|s_1) - P_2^\pi(s_2|s_1) \right| d_0(s_1) \mathrm{d}s_1 \cdots \mathrm{d}s_t \mathrm{d}a_t$$

$$\le R_{\max} \left( \delta_0^t + t\delta_0^{t-1} + \cdots + t\delta_0 \right)$$

$$= R_{\max} \left( (\delta_0 + 1)^t - 1 \right)$$

$$\le R_{\max} \left( (\delta_0 + 1)^H - 1 \right).$$

Noting that $\delta_0 = \tau(\epsilon + M^3 L^{3/2} T\sqrt{dh} + LMmh)\sqrt{T}$, so for $\epsilon$ and $h$ that is sufficiently small, there exists a universal constant $\eta$, such that

$$|\widehat{V}^\pi - V^\pi| \le H\tau H R_{\max} \eta \delta_0 \lesssim R_{\max} \tau^2 H^2 (\epsilon + M^3 L^{3/2} T\sqrt{dh} + LMmh)\sqrt{T},$$

which finishes the proof of Theorem 4.1. $\qquad\square$

## A.3 Auxiliary Lemmas

In this section, we presents the definitions and auxiliary lemmas which are used to prove Theorem 4.2.

**Definition 1** A local martingale $(L_t)_{t \in [0,T]}$ is a stochastic process such that there exists a sequence of non-decreasing stopping times $T_n \to T$ such that $L^n = (L_{t \wedge T_n})_{t \in [0,T]}$ is a martingale.

**Lemma 2**(Chen et al. (2023a), Lemma 16) Let $0 < \zeta < 1$. Suppose that $\mathbf{M}_0, \mathbf{M}_1 \in \mathbb{R}^{2d \times 2d}$ are two matrices, where $\mathbf{M}_1$ is symmetric. Also, assume that $\|\mathbf{M}_0 - \mathbf{I}_{2d}\|_{op} \le \zeta$, so that $\mathbf{M}_0$ is invertible. Let $\mathbf{q} = \exp(-\mathbf{H})$ be a probability density on $\mathbb{R}^{2d}$ such that $\nabla \mathbf{H}$ is $L$-lipschitz with $L \le \frac{1}{4\|\mathbf{M}_1\|_{op}}$, it holds that

$$\left\| \nabla \log \frac{(\mathbf{M}_0)_\# \mathbf{q} * \mathcal{N}(0, \mathbf{M}_1)}{\mathbf{q}}(\theta) \right\| \lesssim L\sqrt{\|\mathbf{M}_1\|_{op} d} + L\zeta \|\theta\| + (\zeta + L\|\mathbf{M}_1\|_{op}) \|\nabla \mathbf{H}(\theta)\|.$$

The following lemmas are very straightforward, so the proof is omitted.

**Lemma 3** Consider $f_n, f : [0, T] \to \mathbb{R}^d$ s.t. there exists an increasing sequence $(T_n)_{n \in \mathbb{N}} \subseteq [0, T]$ satisfying $T_n \to T$ as $n \to \infty$ and $f_n(t) = f(t)$ for every $t \le T_n$. Then for every $\epsilon > 0$, $f_n \to f$ uniformly over $[0, T - \epsilon]$. In particular, $f_n(\cdot \wedge T - \epsilon) \to f(\cdot \wedge T - \epsilon)$ uniformly over $[0, T]$.

**Lemma 4** Consider $f : [0, T] \to \mathbb{R}^d$ continuous, and $f_\epsilon : [0, T] \to \mathbb{R}^d$ s.t. $f_\epsilon(r) = f(r \wedge (T - \epsilon))$ for $\epsilon > 0$. Then $f_\epsilon \to f$ uniformly over $[0, T]$ as $\epsilon \to 0$.

## A.4 EXPERIMENTS

We have made our code publicly available[1].

---

[1] https://anonymous.4open.science/r/bridge_OPE-302D/

