# OpenReview forum: "Schrodinger Bridge to Bridge Generative Diffusion Method to Off-Policy Evaluation"
_ICLR.cc/2024/Conference — Submitted to ICLR 2024_

### Official Review · Reviewer_ijeG · 2023-10-20

**Soundness:** 3 good
**Presentation:** 2 fair
**Contribution:** 2 fair
**Rating:** 5
**Confidence:** 2

**Summary:**

This paper is focused on a new model-based approach to solve OPE. Specifically, it proposes a way to use the diffusion Schrodinger bridge to estimate the probability transition kernel. A theoretical guarantee on the estimation error of the estimated probability transition kernel in total variation distance is provided. Empirical results are also provided for its comparison with existing baselines.

**Strengths:**

This paper introduces a new perspective for doing RL and OPE with diffusion. This is relevant given the recent popularity of diffusion models. The empirical results are nicely done and a good complement to the theoretical results.

**Weaknesses:**

This paper seems to focus on the theoretical aspect of the proposed approach, but the theoretical results presented in this paper can be further developed, for example, by accounting for the effect of sample size on the final estimation error instead of assuming the statistical error is bounded by $\epsilon$.

In addition, Assumption 6 requires the reward function to be well-estimated in the infinity-norm sense, which can be difficult. Normally, the estimation error is bounded in the $\ell_2$-norm sense.

Furthermore, the writing about the algorithm can be made clearer, and there can be more discussion about the implications of Theorem 4.1 and 4.2. Please see the Questions section for details.

**Questions:**

I have the following questions for the authors:

- I had some trouble understanding how the DSB method described in Section 3.1 is applied to OPE. Section 3.2 is not very clear to me. It'd be nice if the authors could explain what $p_{\mathrm{prior}}$ is in the context of OPE as well as what $p_{\mathrm{obs}}$ is in Algorithm 1? And what is the relationship between $p_{\mathrm{prior}}$ and $p_{\mathrm{obs}}$?

- What is $Q^*$ from Section 3.1 in the context of OPE? Is it the Q-function for some policy? And how exactly do we obtain $p_{\mathrm{data}}$ given $Q^*$ and $p_{\mathrm{prior}}$?

- Could the authors explain how $T$ in the algorithm is chosen in general?

- Is "for $k$ in $1:K$" in Algorithm 1 a typo and what the authors actually meant is "for $k$ in $1:N$"?

- Is gradient descent sufficient for minimizing $\tilde{\mathcal{L}}_{SB}(\mathbf{x}_T; \theta)$ with respect to $\theta$? If not, is a local minimizer $\theta$ sufficient for the theoretical results? Similar question for $\phi$.

---

> ### Author Response · Authors · 2023-11-22
> **Response to the Reviewer ijeG**
>
> > What is $Q*$  from Section 3.1 in the context of OPE? Is it the Q-function for some policy? And how exactly do we obtain $p_{\mathrm{data}}$  given $Q*$ and $p_{\mathrm{prior}}$
>
>  $Q*$ from section 3.1 is not the Q-function for some policy, it is a path measure on $\mathcal{S}$ through time interval $[0,T]$ with marginal densities $Q_0^*=p_{{data}}$ and $Q_T^*=p_{\mathrm{prior}}$, which is the solution of the Schrodinger bridge problem (equation (3)). As a result, given $Q*$  and $p_{\mathrm{prior}}$ we can easily achieve $p_{\mathrm{data}}$ by integrating the path measure.
>
>
>
> > Could the authors explain how in the algorithm is chosen in general?
>
> $T$ in the implementation is always chosen to be 1, and discretized into 50 or 100 time steps, i.e. the stepsize $h=0.02$ or $h=0.01$.
>
> > Is for $k$ in $1:K$ in algorithm 1 a typo and what the authors actually meant for $k$ in $1:N$
>
> Thank the reviewer for bringing this up. This is not a typo and $K$ represents the number of stages in the training procedure. In one stage, we train the forward and backward process by multiple gradient descent.  We need a repetitive training process, continuously updating the corresponding backward diffusion process through the well-trained forward diffusion process and vice versa.

---

### Official Review · Reviewer_p1r4 · 2023-10-30

**Soundness:** 2 fair
**Presentation:** 2 fair
**Contribution:** 1 poor
**Rating:** 3
**Confidence:** 4

**Summary:**

The paper studies off-policy evaluation in reinforcement learning, specialized to an episodic MDP setting. The method is simple to describe. It employs the conditional diffusion Schrodinger bridge estimator to learn the Markov transition kernel and then computes a model-based policy value estimator based on the estimated transition kernel. The author(s) further established an upper error bound for the policy value estimator. Numerical experiments were further conducted to investigate its finite sample performance.

**Strengths:**

The merits of the paper can be summarized as follows:

* The paper appears to be the first work in introducing the diffusion Schrödinger bridge generative model to the domain of off-policy evaluation, to my best knowledge.

* The initial sections of the document are presented with great clarity, making them accessible and straightforward for the reader to grasp the core concept.

* Through a series of numerical experiments, the paper successfully demonstrates the superior performance of the proposed method over existing solutions in some of the selected datasets, demonstrating its potential.

**Weaknesses:**

The paper exhibits several weaknesses that warrant careful consideration and potential revision:

* **Novelty**: The method presented seems to be an amalgamation of the pre-existing condition diffusion Schrödinger bridge estimator, as outlined by Chen et al. (2023), and the model-based off-policy evaluation. However, the paper falls short in articulating the rationale behind this integration. Specifically, it does not sufficiently illuminate the advantages of employing the condition diffusion Schrödinger bridge estimator. Comparatively, in the existing literature, it is a common approach to use a Gaussian dynamics model with parameters defined via deep neural networks for estimating conditional density functions, which has demonstrated faster estimation and inference processes, as well as practical efficiency (see, for example, [Paper 1](https://arxiv.org/pdf/2005.13239.pdf), [Paper 2](https://proceedings.neurips.cc/paper/2020/file/f7efa4f864ae9b88d43527f4b14f750f-Paper.pdf), [Paper 3](https://arxiv.org/pdf/2106.03207.pdf), [Paper 4](https://arxiv.org/pdf/2301.02220.pdf)). Moreover, in the numerical experiments, there are instances where the proposed method results in significantly larger errors compared to Fitted Q-Evaluation (FQE) or model-based (MB) approaches.

*  **Familiarity with Off-Policy Evaluation (OPE) Literature**: The author(s) appears to have a limited understanding of the OPE literature. Certain claims made in the paper might be inaccurate. For example, previous works such as Jiang & Li (2016), Precup et al. (2000), Thomas et al. (2015), and Thomas & Brunskill (2016) actually employed the sequential importance sampling (IS) ratio instead of the marginal importance sampling ratio for OPE. The paper attributes the marginal IS ratio to these works in the second paragraph of the introduction. The correct attribution should be made to Liu et al. (2018) Paper 5, which was the first to propose the use of the marginal importance sampling ratio, with subsequent developments in the DICE-type estimators (e.g.,  https://arxiv.org/abs/2010.11652) and other extensions (e.g., https://arxiv.org/pdf/1909.05850.pdf, https://arxiv.org/pdf/2109.04640.pdf and https://proceedings.mlr.press/v139/shi21d/shi21d.pdf). Additionally, Liu et al. (2018) also explored minimax optimization for computing the marginal IS ratio. Furthermore, it was mentioned on Page 2 that "The idea of using generative model as transition function estimator in RL, to our knowledge, has not been discovered in the literature". However, as commented in my previous comment, there have been several works in using Gaussian generative models for policy learning.

* **Clarity of Sections 3 and 4**: These sections are challenging to follow and would benefit significantly from a simplification of notation and presentation to enhance readability and comprehension.

* **Theoretical Analysis and Coverage Assumption**: The theoretical analysis is missing a discussion on the crucial coverage assumption, which necessitates that the ratio between the behavior and target policy be finite. This assumption is foundational for offline RL. Unfortunately, the paper does not provide assumptions or explanations on this aspect, leaving a gap in the theoretical groundwork.

* **Inadequacy of Numerical Analysis**: The numerical analysis conducted seems insufficient. The paper explores only four benchmark environments with a fixed number of episodes. Additionally, the numerical experiments consider infinite horizons, whereas the paper focuses on fixed episodic settings. Moreover, the proposed estimator demonstrated larger errors in comparison to FQE or MB in two of the environments, questioning its practical effectiveness. To provide a more comprehensive evaluation, it would be beneficial to vary the sample size and offer a thorough comparison across different settings.

**Questions:**

Do you need the coverage assumption to guarantee the consistency of the estimated Markov transition kernel?

---

> ### Author Response · Authors · 2023-11-22
> **Response to the Reviewer p1r4**
>
> > The numerical analysis conducted seems insufficient. ... To provide a more comprehensive evaluation, it would be beneficial to vary the sample size and offer a thorough comparison across different settings.
>
> We thank the reviewer for bringing it up.  We have added experiments demonstrating the empirical performance of our estimator is consistent with our claim.  We refer to our rebuttal for reviewer VmGz for the details of the additional experiments.
>
> > Do you need the coverage assumption to guarantee the consistency of the estimated Markov transition kernel?
>
> We do not need to additionally state the coverage assumption in the paper, since the coverage assumption is already contained in assumption 6 in our paper. For example, if $\mu$ is degenerate and the target policy $\pi$ is non-degenerate, there will be data points $(s,a)$ that are included in $\pi$'s support range but don't appear in the dataset $\mu$ generated (since they are not included in $\mu$'s support range), which we use to estimate our score. As a result, the estimation error
>
> $\mathbb{E}_{q}[\| \mathbf{Z}(kh,\mathbf{X},(\theta,t,s,a))-\mathbf{Z} \|^2]$
>
> on these $(s,a)$ will be a constant instead of $\epsilon^2$, and assumption 6 does not hold. By the above argument, it requires that the support of $\pi$ to be a subset of the support of $\mu$ (that is, $\sup_{s,a}\frac{\pi(a|s)}{\mu(a|s)}<\infty$, i.e.the coverage assumption) for assumption 6 to hold.
>
> > I had some trouble understanding how the DSB method described in Section 3.1 is applied to OPE. Section 3.2 is not very clear to me. It'd be nice if the authors could explain what $p_{\mathrm{prior}}$  is in the context of OPE as well as what $p_{\mathrm{obs}}$  is in Algorithm 1? And what is the relationship between $p_{\mathrm{prior}}$ and $p_{\mathrm{obs}}$?
>
> $p_{prior}$ is the prior distribution in diffusion process, and usually it is a standard gaussian distribution. $p_{obs}$ is the same as $p_{data}$ and we apologize for using this term without defining it first. We shall make our notation more consistent in a revised paper.

---

### Official Review · Reviewer_VmGz · 2023-10-31

**Soundness:** 3 good
**Presentation:** 2 fair
**Contribution:** 1 poor
**Rating:** 3
**Confidence:** 3

**Summary:**

The authors consider the problem of off policy evaluation. They attempt to learn an estimator for V, the value function, by learning an estimator for the reward function R and the transition kernel T. The former they model as a neural network, while for the latter they resort to a conditional diffusion Schrödinger bridge. The authors ultimately end up with a function T, which is conditioned on the timestep, with which they proceed to model V. They investigate the performance of their algorithm theoretically, and with experiments.

**Strengths:**

- The authors consider a problem that is both contemporary and valuable
 - The authors provide detailed theoretical analysis of their contribution

**Weaknesses:**

- The core methodology presented in the paper is prior art (Chen2021, Chen2023b, Chen2023c)
 - The experimental evaluation is extremely limited
 - The few experimental results that are given do not demonstrate a performant method

**Questions:**

- Can the authors elaborate why the performance of their method is so inconsistent in experiments? A clear trend (as theretical results would suggest) is absent.
 - The core of the contribution seems to be section 3.3, but there are a few elements that are not quite clear
   * Why does the loss need to be masked?
   * What does it mean to add the time to the training parameters? Does this mean time gets gradient updates in every step? This seems like a mistake.
   * Overall, this section could do with a careful rewrite, in particular since this section seems to be the core part of the methodological innovation in the paper
 - Can the authors explain how the integral in (1) is evaluated in practice? While the paper discusses how to achieve the individual factors in the integrand, they do not discuss how they evaluate the integral in practice? Imortance sampling? SMC?
 - While the paper is very rigorous in defining the assumptions of the theoretical analysis (which is a big plus!), it is not clear how feasible each of these is in practice. In particular, do the authors have any means of evaluating whether these requirements are indeed satisfied in their trained estimators?
Small notes:
 - There is a latex typo in eqn 5 "Psi"
 - Is the data in figure 1 identical to that in table 1?
Justification for score:
 Overall I think the experimental evaluation is too limited, and the existing experimental evaluation is not in the authors' favor. I also think there are certain obvious aspects of the experiments that are missing In particular, if theoretical results indicate favorable results with respect to horizon length, I would expect some result that indeed shows this scaling, and how it improves compared to other methods. This would require at least giving the expected horizon length, rather than simply labelling it "infinite". The same is true for action and state space dimension. The authors demonstrate favorable scaling, but no thorough investigation is done to confirm this, or show that other methods do not have this favorable scaling.

---

> ### Author Response · Authors · 2023-11-22
> **Response to the Reviewer VmGz**
>
> > The core of the contribution seems to be section 3.3, but there are a few elements that are not quite clear
> >
> > * Why does the loss need to be masked?
> > * What does it mean to add the time to the training parameters? Does this mean time gets gradient updates in every step? This seems like a mistake.
> > * Overall, this section could do with a careful rewrite, in particular since this section seems to be the core part of the methodological innovation in the paper
>
> We thank the reviewer for bringing those questions up. During our training, the time of the MDP $t$, action $a_t$,  current and next state $s_{t-1}$ and $s_t$ are all inputs of the diffusion direction $Z(r, \mathbf{X}_r,\mathbf{M},\phi)$ parameterized by a neural network with parameters $\phi$. We apologize for the typos and misleading expressions in our writing. We will revise this part for better readability.
>
>
>
> > Can the authors explain how the integral in (1) is evaluated in practice? While the paper discusses how to achieve the individual factors in the integrand, they do not discuss how they evaluate the integral in practice? Imortance sampling? SMC?
>
> We thank the reviewer for bringing it up. The integral in (1) is evaluated using Monte Carlo method as we outlined in *Model-based OPE* of Algorithm 1. To evaluate the expected reward, we need to sample from the target policy that is given and the transition which is learned using the Schrodinger Bridge diffusion model on offline data. Then we calculate the reward based on the trajectories we sampled from the learned environment.
>
> > Is the data in figure 1 identical to that in table 1? ....The authors demonstrate favorable scaling, but no thorough investigation is done to confirm this, or show that other methods do not have this favorable scaling.
>
> We thank the reviewer for bringing it up. To further demonstrate the empirical behavior of our method, we carry out experiments investigating how error change with horizon and number of trajectories. The existing experiments is carried out on a benchmark proposed by Fu et.al.[1], which only provides settings for infinite dimensional MDP and the number of trajectories in the behavior dataset also cannot be changed. Therefore, we now utilize the benchmark in Voloshin et.al.[2] to verify our results.
>
> This time, we do additional experiments on pixel Gridworld which is a pixel-based Grildworld with state being 64-dimensional. We compare it with Model-Based Method and Direct Method Regression meanwhile validating the empirical behavior is in favor of our major theoretical claim. These two baselines are provided in the open-source implementation of the second benchmark. We show the result in two graphs,  [graph1](https://ibb.co/3BtQvzv), [graph 2](https://ibb.co/bB5v9G7 ).
>
> From the second plot, we can see a polynomial dependence on horizon of the absolute error, which is consistent with the result in our theoretical claim.
>
> > While the paper is very rigorous in defining the assumptions of the theoretical analysis, it is not clear how feasible each of these is in practice. In particular, do the authors have any means of evaluating whether these requirements are indeed satisfied in their trained estimators?
>
> We thank the reviewer for pointing out this gap. Assumptions 2 to 5 are easily feasible and can be straightforwardly checked. Assumption 1 is the Lipschitz continuity of the score function, which is also a common assumption in diffusion modeling which covers a wide range of problems. Papers such as [3] and [4] use similar assumpition. Assumption 6 is also a common learning error assumption which can be obtained via standard stastical analysis, as is stated in [5], [6] and [7]. We will include these justifications in our revised paper.
>
> [1] Fu, Justin, et al. "Benchmarks for Deep Off-Policy Evaluation." *International Conference on Learning Representations*. 2020.
>
> [2] Voloshin, Cameron, et al. "Empirical study of off-policy policy evaluation for reinforcement learning." *arXiv preprint arXiv:1911.06854* (2019).
>
> [3]Sitan Chen, Sinho Chewi, Jerry Li, Yuanzhi Li, Adil Salim, and Anru R. Zhang. Sampling is as easy
> as learning the score: theory for diffusion models with minimal data assumptions, 2023a.
>
> [4]Chen, H., Lee, H. \& Lu, J.. (2023). Improved Analysis of Score-based Generative Modeling: User-Friendly Bounds under Minimal Smoothness Assumptions. Proceedings of the 40th International Conference on Machine Learning,
>
> [5]Block Adam, Mroueh Youssef, Rakhlin Alexander. (2020). Generative Modeling with Denoising Auto-Encoders and Langevin Sampling.
>
> [6]Xiuyuan Cheng, Jianfeng Lu, Yixin Tan, and Yao Xie. Convergence of flow-based generative models
> via proximal gradient descent in wasserstein space, 2023.
>
> [7]Gen Li, Yuting Wei, Yuxin Chen, and Yuejie Chi. Towards faster non-asymptotic convergence for
> diffusion-based generative models, 2023.

---

> > ### Comment · Reviewer_VmGz · 2023-11-22
> >
> > I thank the authors for responding to me, but I will keep my score as is, my reasoning is as follows:
> >  - The authors have not provided an answer to my main question about lacking performance
> >  - The experimental validation in the paper is very minimal, and as pointed out, the results that are there are not convincing.
> >  - The updated experimental results further support the idea that the method does not seem to perform well in practice. The first graph shows that their method performs worst among the three methods. The second graph shows that while the scaling indeed looks polynomial with episode length, the baselines are not shown for comparison in that graph, so it is not clear this is an advantage over other methods.

---

### Official Review · Reviewer_CNAs · 2023-11-01

**Soundness:** 2 fair
**Presentation:** 2 fair
**Contribution:** 1 poor
**Rating:** 3
**Confidence:** 3

**Summary:**

The authors presents the novel model-based off-policy evaluation method based on diffusion Schrodinger bridge generative model. The established statistical rate shows only polynomial dependence on planning horizon, thus improves over classic importance sampling methods. Finally, authors present numerical validation of their method on several continuous RL environments.

**Strengths:**

- First application of diffusion models to off-policy evaluation problem;
- Theoretical guarantees that avoid the curse of horizon and dimensionality;

**Weaknesses:**

- Theoretical contribution is very limited.
    - Assumption 6 in Section 4 is very strong because it requires very good exploration properties of the behavior policy: there should be enough data for any state-action pair to learn the model uniformly for all state-action pairs.
    - Additionally, it is very unlikely that the estimation error is available for practical implementations since this error defined in terms of integrals with respect to the optimal Schrodinger bridge, and also uniform over all state-action pairs.
    - Corollary 5 of (Liu et al. 2020) yields that in the case of existence of a good approximation of importance weights (no matter the way of obtaining it), the approximate SIS algorithm also have polynomial dependence in horizon. As a result, the presented method is not the first implementable method that avoids exponential dependence on the horizon.
        - Liu, Y., Bacon, P. L., & Brunskill, E. (2020, November).
        Understanding the curse of horizon in off-policy evaluation via
        conditional importance sampling. In *International Conference on Machine Learning* (pp. 6184-6193). PMLR.
    - Theorem 4.2 holds only for DDPM-type models and repeats the proof of Theorem 10 by (Chen et al, 2023a) line-by-line
        - Sitan Chen, Sinho Chewi, Jerry Li, Yuanzhi Li, Adil Salim, and Anru R. Zhang. Sampling is as easy as learning the score: theory for diffusion models with minimal data assumptions, 2023a, ICLR 2023
    - Proof of Theorem 4.1 might be very simplified by using Bellman equations and backward induction. In this case it becomes a very simple corollary of Theorem 4.2.
- Regarding experimental validation, there is a lack of comparison with importance sampling and doubly-robust methods.
- Application of diffusion model to RL in general is not novel and there is several method that apply diffusion to offline RL problem, e.g.
    - Wang, Z., Hunt, J. J., & Zhou, M. (2022). Diffusion policies as an expressive policy class for offline reinforcement learning. ICLR 2023

**Questions:**

- No direct assumptions on $\mu$ looks very strange for off-policy evaluation algorithms since in the worst case $\mu$ may be degenerate distribution whereas the goal is to evaluation non-degenerate one; It requires additional comments.
- Details on neural network, training procedure, inference baseline models and evaluation are not in appendix at it is written in the main text, there is only a link to a code. Is it possible to provide these details?
- Additionally, the guarantees claimed in the abstract are very confusing since the policy error is not decreasing with any parameters of the method, whereas Theorem 4.1 shows that error decreases with approximation error and discretization step of SDE.

Misprints and confusing or undefined notaiton

- Definition of transition kernel in the beginning of Section 2: current definition by **probability** of transition implies that a support of $T(s,a)$ is at most countable since the probability should be summable to $1$. Should be it defined as a probability density function?
- What is expectation of the **event**? What is conditional expectation of the event?
- Misprint in a definition of a density of a push-forward measure: $f$#$q$ with undefined $q$;
- Missed $\mathrm{d} s_{t-1}$ in the integral that defines $d^pi_t(s_t)$ on page 3;
- Equation (5): there is a misprint in Psi;
- Excess comma after equation (10);
- End of page 7: “lipschitzness” instead of “Lipschitzness”;
- 6th line after Theorem 4.1: “Finally, The” instead of “Finally, the”;
- Lemma 1:  “where” is written combined with the next part of the formula;
- Q-martingales and Q-Brownian motion are not defined in Lemma 1.
- Start of page 15: misprint in Psi.

---

> ### Author Response · Authors · 2023-11-22
> **Response to the Reviewer CNAs**
>
> > No direct assumptions on $\mu$ looks very strange for off-policy evaluation algorithms since in the worst-case may be degenerate distribution whereas the goal is to evaluation non-degenerate one; It requires additional comments.
>
>  We thank the reviewer for pointing it out, for it does need clarification. In fact, although we don't have direct assumptions on $\mu$, assumption 6 requires good properties of $\mu$ to achieve the small estimation error of the scores. For example, if $\mu$ is degenerate and the target policy $\pi$ is non-degenerate, there will be data points $(s,a)$ that are included in $\pi$'s support range but don't appear in the dataset $\mu$ generated which we use to estimate our score. As a result, the estimation error
>
> $\mathbb{E}_{q}[||Z(kh,X,(\theta,t,s,a))-Z||^2]$
>
> on these $(s,a)$ will be a constant instead of $\epsilon^2$, and assumption 6 does not hold. By the above argument, it requires that the support of $\pi$ to be a subset of the support of $\mu$ (that is, $\sup_{s,a}\frac{\pi(a|s)}{\mu(a|s)}<\infty$) for assumption 6 to hold. In conclusion, the potential assumptions on $\mu$ for our theorem to hold is contained in assumption 6 which we already stated in our paper.
>
>
>
> > Details on neural network, training procedure, inference baseline models and evaluation are not in appendix at it is written in the main text, there is only a link to a code. Is it possible to provide these details?
>
> We thank the reviewer for bringing it up. Those details will be added to the revised paper.
>
> * For the neural network, we use a modified U-Net with timestep embedding for the forward diffusion direction $Z(r, \mathbf{X}_r,\mathbf{M},\phi)$ and a transformer for the backward direction $\hat Z(r, \mathbf{X}_r,\mathbf{M},\phi)$. We refer to section c.4 in the appendix of the work by Chen et al.[1] for details.
> * For the training procedure, we will add all the hyperparameters, including batch size, discretization, and optimizer in the revised paper.
> * For the baseline and evaluation, we have fully followed their implementation as described in the work of Fu et al.[2], where an open-source implementation is available.
>
> > Additionally, the guarantees claimed in the abstract are very confusing since the policy error is not decreasing with any parameters of the method, whereas Theorem 4.1 shows that error decreases with approximation error and discretization step of SDE.
>
> We thank the reviewer for bringing up their confusion. The conclusion of our theoretical analysis is that when the approximation error is small, good OPE result can be obtained using our method without considering the dimension and horizon factor, which avoids the case where the OPE error grows exponentially with respect to dimension and horizon length which many other methods suffer from. We will rephrase our abstract and introduction more adequately to avoid confusion.
>
>
>
> > Definition of transition kernel in the beginning of Section 2: current definition by probability of transition implies that a support of $T(s,a)$ is at most countable since the probability should be summable to 1. Should be it defined as a probability density function? What is expectation of the event? What is conditional expectation of the event?
>
> The transition kernel $T(s,a)$ is indeed defined as a probability density function, we are sorry for the confusion of definition. The expectation and conditional expectation of an event are samely defined as in probability theory. We will clear the typo and confusing notation in our revised paper.
>
> [1] Chen, Yu, et al. "Provably Convergent Schr\" odinger Bridge with Applications to Probabilistic Time Series Imputation." *arXiv preprint arXiv:2305.07247* (2023).
>
> [2] Fu, Justin, et al. "Benchmarks for Deep Off-Policy Evaluation." *International Conference on Learning Representations*. 2020.

---

### Meta-Review · Area_Chair_CGF8 · 2023-12-12

**Metareview:**

In this paper, the authors introduce Schordinger bridge for model-based off-policy evaluation. Based on the learned diffusion model, a Monte-Carlo approximation is used for policy evaluation upon the learned model.

**Justification For Why Not Higher Score:**

All the reviewers believe this paper is not ready to be published:

1, Theoretical contribution is very limited. The diffusion model learning is repeated from Chen et al, 2023a. The claim about the disadvantage of MIS vs. advantage of model-based OPE is not valid. In fact, there will be distribution shift, which may lead to distribution coverage issue in OPE, has not been discussed.

2, Empirical comparison is limited, without the comparison w.r.t. MIS and DICE algorithms.

**Justification For Why Not Lower Score:**

N/A

---

### Decision · Program_Chairs · 2024-01-16

Reject